# Correlation between the Desiccator Method and 1 m³ Climate Chamber Method for Measuring Formaldehyde Emissions from Veneered Particleboard

Jia Shao [1,2], Yang Chen [1], Ling Dong [2], Tangfeng Yuan [2], Zhongfeng Zhang [1] and Jijuan Zhang [1,*]

[1] College of Furniture and Art Design, Central South University of Forestry and Technology, Changsha 410004, China; 20211200365@csuft.edu.cn or richardshaojia@163.com (J.S.); 20191200262@csuft.edu.cn (Y.C.); t19990735@csuft.edu.cn (Z.Z.)

[2] Oppein Home Group INC, Guangzhou 510000, China; dongling@oppeinmail.com (L.D.); yuantangfeng@oppeinmail.com (T.Y.)

\* Correspondence: t20050729@csuft.edu.cn; Tel.: +86-139-7511-9348

**Abstract:** To shorten the measuring time of formaldehyde emissions from wood-based panels and reduce the costs of quality control processes during industrial furniture production, more efficient methods for measuring formaldehyde emissions from wood-based panels need to be developed. In this study, the formaldehyde emissions from 18-mm-thick veneered particleboard were measured using the desiccator method and the 1 m³ climate chamber method according to Chinese national standard GB/T17657-2013, and the correlation between these two methods was determined. Through a correlation analysis of 60 groups of data, the results indicated that the linear correlation coefficient (R) between two methods was 0.718, and the regression model was established, which by F and P values demonstrated a significant correlation at the 0.01 level of significance. As long as the quality of materials and the production processes remained consistent, the desiccator method was reliable enough for conducting routine quality control measurements of formaldehyde emissions from veneered boards of 18 mm thickness. In case of dispute, the results can be verified using the 1 m³ climate chamber method for accuracy.

**Keywords:** desiccator method; climate chamber method; formaldehyde emissions; veneered particleboard; wood-based panels; correlation coefficient

## 1. Introduction

Panel-type furniture enterprises require a substantial amount of wood-based panels, such as particleboard, fiberboard, and plywood, as raw materials. In 2020, the total output of wood-based panels in China was over 300 million m³, and the scale of China's wooden furniture market exceeded 600 billion yuan. Among them, solid wood furniture accounted for 41.8%; panel furniture accounted for 58.2%. These wood-based composites are usually bonded with formaldehyde-based adhesives, such as urea-formaldehyde resin, melamine-formaldehyde resin, and phenolic resin, which inevitably leads to formaldehyde release into the surrounding indoor environments [1]. Formaldehyde release from wood-based panels is a complicated process, which can be affected by several factors: the physical and chemical properties of the materials, such as formaldehyde content, component structure, chemical composition, density, thickness, and surface properties of the material; and environmental factors, such as temperature, relative humidity, air velocity, and air exchange rate [2–4]. Honggang Chen found that a high concentration of formaldehyde is toxic to the nervous system, immune system, and liver; it is carcinogenic to humans [5]. Formaldehyde has been classified as a potentially dangerous carcinogen and an important environmental pollutant by the World Health Organization and the United States Environmental Protection Agency. The testing methods of formaldehyde can be summarized in three categories: total amount

testing method, such as perforation; static emission testing method, such as desiccator; and dynamic emission testing method, such as chamber. The chamber method is widely used in the United States and Germany, and its testing technology of chamber is in a leading position in the world. The EU standards use chamber, gas analysis, and perforation methods. Japan is using the desiccator method [6–10]. The Chinese standard GB18580-2017 [11] stipulates that the measurement of formaldehyde emissions from wood-based panels shall be carried out in accordance with the 1 m$^3$ climate chamber method in the GB/T 17657-2013 standard, and the desiccator method, the perforator method, is for production quality control in enterprise [12]. The formaldehyde emission standards of wood-based panels in different countries are shown in Table 1 [13].

**Table 1.** Formaldehyde emission standards for wood-based panels in Europe, the USA, Japan, Australia, and China.

| Country | Standard | Test Method | Board Class | Limit Value |
|---|---|---|---|---|
| Europe | EN13986: 2005 | Perforator EN ISO 12460-5 | E1-unfaced particleboard, MDF/HDF, OSB | ≤8 mg/100 g * |
| | | Chamber EN 717-1 | E1-particleboard, MDF/HDF, OSB | ≤0.1 ppm ** |
| | | Gas analysis EN 717-2 | E1-unfaced plywood, solid wood panels, laminated veneer lumber (LVL) | ≤3.5 mg/m$^2$ h |
| | | Gas analysis EN 717-2 | E1-coated, overlaid, or veneered particleboard, OSB, fibreboard, plywood, solid wood panels, LVL, cement-bonded particleboard | ≤3.5 mg/m$^2$ h |
| USA | ANSI A 208.1 & 2 | ASTM E1333 (chamber) | Particleboard/MDF | ≤0.18 or 0.09 ppm/≤ 0.21 or 0.11 ppm |
| Japan | JIS A 5908 (2015) and 5905 | JIS A 1460 (Desiccator) | F **/F ***(E0)/ F ****(SE0) | ≤1.5 mg/L/ ≤0.5 mg/L/ ≤0.3 mg/L |
| Australia and New Zealand | AS/NZS 1859/1 (2017) and 2 | AS/NZS 4266.16 (Desiccator) | E0-particleboard, MDF/E1-particleboard /E1/MDF | ≤0.5 mg/L/ ≤1.5 mg/L/ ≤1.0 mg/L |
| China | GB18580-2017 | GB/T 17657-2013 (chamber) | E1-MDF, particleboard, plywood, LVL, or veneered wood-based panel | ≤0.124 mg/m$^3$ |

* E3 30–60 mg/100 g, E2 8–30 mg/100 g, E1 5–8 mg/100 g, E0 ≤ 3 mg/100 g, super E0 ≤ 1.5 mg/100 g. ** 0.05 ppm boards can be marked with an environmental label ("Blue Angel"), 0.03 ppm boards are about equal to the Japanese emission class F ****.

The climate chamber method considers the temperature, relative humidity, loading ratio, air exchange rate, and air speed on the sample surface; therefore, it most closely reflects real-world conditions. However, the climate chamber method is not suitable for routine enterprise production quality control because it is time consuming. The desiccator method is an internationally recognized method because it enables a much faster measurement of formaldehyde emissions from wood-based panels, and it is inexpensive and relatively easy to conduct, which has made it popular in Chinese furniture enterprises. This study sought to develop a relevant mathematical model for establishing a relationship between the desiccator method and the 1 m$^3$ climate chamber method. This model would make it convenient for enterprises to adopt desiccator methods for production quality control and to meet the requirements of the GB18580-2017 standard.

A significant number of experimental studies have been conducted to establish relationships between different formaldehyde-measuring methods. Chris Leffel determined the empirical equation y = 29.332x + 4.2569 to convert between the measured values obtained from the perforator method (y) and the American large climate chamber method (x) [14]. Yongliang Chi (2015) measured the formaldehyde emission of medium-density fiberboard (MDF) using the 9–11 L desiccator (A) and perforator (D) methods, and the statistical analysis results showed that there was a linear relationship between the two methods. The linear regression equation was D = 4.8953A + 2.3412, and its correlation coefficient ($R^2$) was 0.9960, indicating a significant correlation [15]. Yiqing Peng measured the formaldehyde emission of MDF using the 40 L desiccator, gas analysis, and perforator methods. Upon analysis of the results using SPSS software, a relevant regression model and linear equation were established, the results of which demonstrated a highly positive correlation between the three measurement methods [16]. Qionghui Zhao measured the formaldehyde emission of blockboard by two different methods, the desiccator method and the 1 $m^3$ climate chamber method and compared the results to elucidate the influence of different measurement methods on the results of measurement of formaldehyde emissions. The results showed that the two methods were consistent in the grade characterization of the panels, and there was a correlation between the standard curves of the two methods [17]. Xiaorong Lin and Xiyuan Liang took 20 different wood-based panel samples with different specifications and models, and the formaldehyde emissions from each sample were measured using the 1 $m^3$ climate chamber method and the desiccator method. Through comparative analysis, the formaldehyde emissions from the MDF, particleboard, and plywood samples measured using the 1 $m^3$ climate chamber method were in the range of 0.100–0.124 mg/$m^3$, while the emissions measured using the desiccator method were in the range of 0.5–0.7 mg/L. When the concentration of formaldehyde measured using the desiccator method reached 0.7 mg/L, the concentration measured using the climate chamber method was below the limit of 0.124 mg/$m^3$ [18]. These studies elucidated the correlations between the different formaldehyde measuring methods and provided an impetus for developing reliable and feasible quality control methods for the measurement of formaldehyde emissions from wood-based panels. In addition, the identification and analysis of the factors influencing the accuracy and reliability of different measurement methods could provide guidance to ensure the accuracy of the data obtained in this study.

Other studies have reported poor correlations between different methods of measuring formaldehyde emission. Previously, Jimei Wang and Zhijiang Ji conducted a comparative analysis on seven samples of plywood, particleboard, and blockboard, the results of which indicated that the concentration of formaldehyde emitted from the three types of wood-based panels differed depending on the method of measurement (i.e., perforator method, desiccator method, and climate chamber method); thus, there was no correlation or comparability between them [19]. The researchers attributed the lack of correlation to the poor uniformity of the wood-based panels themselves and to the differences in the production process at each manufacturer, making it difficult to conduct a unified correlation analysis. In addition, the data gathered were insufficient to accurately demonstrate a correlation between the results. Lastly, errors were inevitable because of the different measuring conditions of the enterprises, operation methods of operators, and other factors.

Since the implementation of the new Chinese national standard, the methods available to measure formaldehyde emissions have undergone continuous development and improvement in practice. To guide the formaldehyde measurement and quality control of wood-based panels, it was necessary to study the correlation between different measurement methods. However, all of the current studies were limited to a small range of laboratory-scale analyses and comparisons; most studies comprised only 15–30 groups of experiments. Thus, the amount of experimental data was not comprehensive enough to draw conclusions for establishing a reliable method for measuring formaldehyde emissions in industrial production. In this study, we increased the number of standardized measurements to 60 groups. We hypothesized that these additional data would make

the correlation studies more robust and provide a reliable and accurate reference for the rapid measurement of formaldehyde emissions for quality control during actual industrial production processes.

## 2. Materials and Methods

### 2.1. Materials

A total of 60 pieces of veneered particleboard were obtained from Oppein Home Group, Inc. The dimensions of the boards were 1220 mm × 2440 mm × 18 mm.

#### 2.1.1. Sample Preparation

The 60 sample groups were numbered from 1 to 60. Each specimen was cut into two pieces with dimensions of 1220 mm × 1200 mm, as shown in the sample crosscut diagram in Figure 1. The pieces of each specimen were marked as X-1 and X-2, respectively (i.e., 1-1, 1-2, 2-1, 2-2, and so on), and the samples were wrapped with a plastic film that did not adsorb or release formaldehyde to be measured.

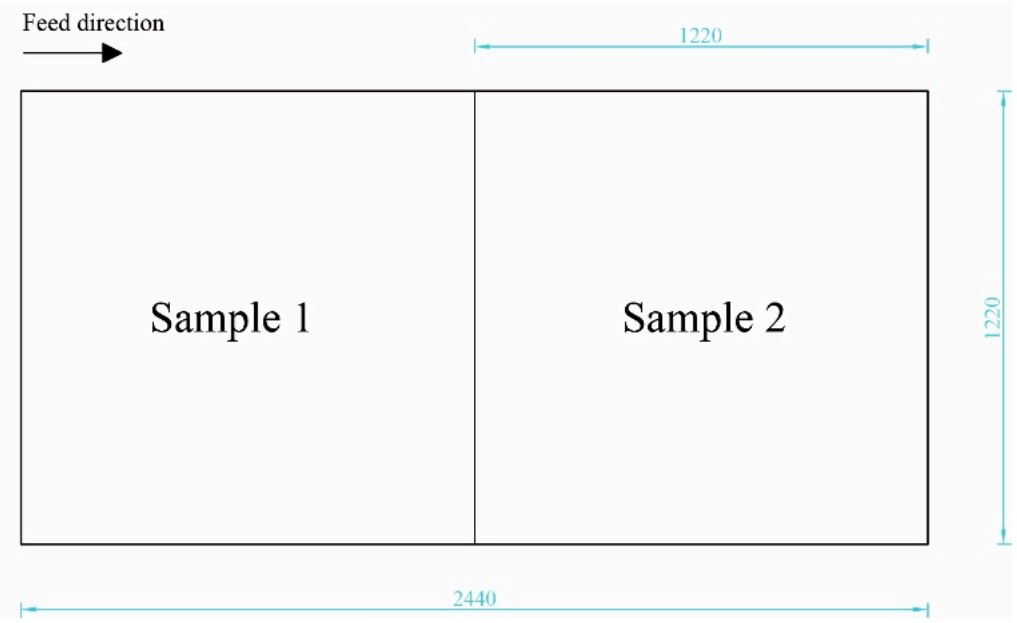

**Figure 1.** Sample preparation.

The samples whose formaldehyde emissions were measured using the 1 m³ climate chamber method of GB/T17657-2013 comprised group A. Two pieces with dimensions of 500 mm × 500 mm × 18 mm were made, respectively from sample 1 and sample 2 according to the positions shown in Figures 2 and 3. The samples whose formaldehyde emissions were measured using the desiccator method of GB/T17657-2013 comprised group B. Ten pieces with dimensions of 150 mm × 50 mm × 18 mm were prepared from sample 1 and sample 2 according to the positions shown in Figures 2 and 3. An additional standby group was prepared, totaling 30 pieces. For example, the samples for the 1 m³ climate chamber method were labeled 1-A1 and 1-A2, and the samples for the desiccator method were labeled 1-B1, 1-B2, and 1-B3. The other groups were named in the same way. The cutting edge and ends of each sample needed to be at least 50 mm from the plate edge.

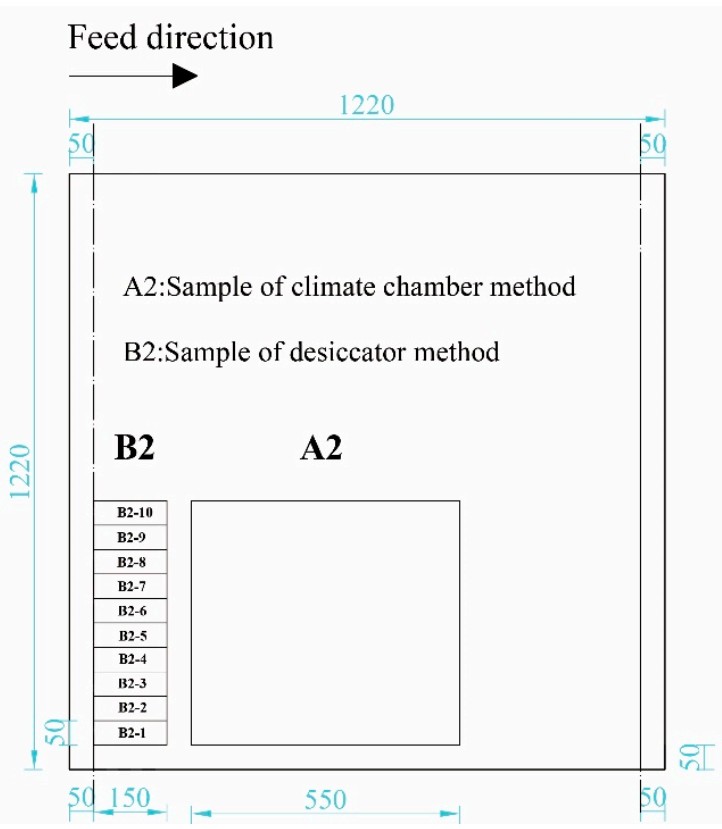

**Figure 2.** Sampling diagram of sample 1.

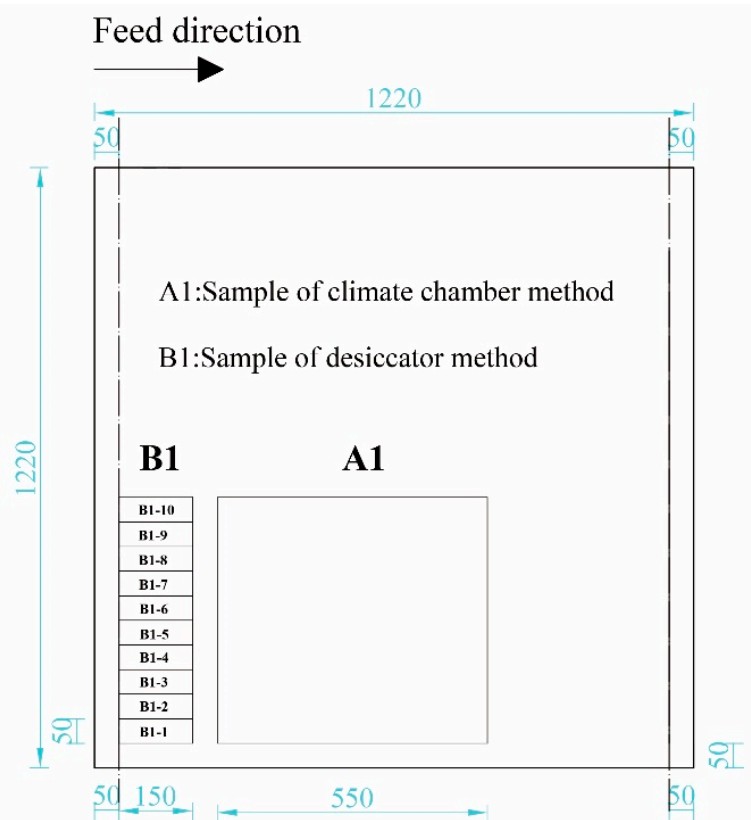

**Figure 3.** Sampling diagram of sample 2.

### 2.1.2. Reagents

Acetyl acetone (analytical purity), ammonium acetate (analytical purity), glacial acetic acid (analytical purity), and 10.1% formaldehyde standard solution ($CH_2O$).

### 2.2. Instruments and Equipment

The main equipment used in this study included a QWH-1000C 1 $m^3$ climate chamber (Hainate, Jinan, China), TY-210-4 formaldehyde balance chamber (Mingchi, Dongguan, China), UV-1500 UV spectrophotometer (ShouKe, Guangzhou, China), 723PC visible spectrophotometer (Shunyao Hengping, Shanghai, China), Model 501 constant-temperature water boiler (Aohua, Changzhou, China), QC-3 atmospheric sampler (Lubo, Qingdao, China), 608-H1 temperature and humidity detector (Testo, Titisee-Neustadt, Germany), and desiccator (diameter: 240 mm, volume: 9–11 L) (Lubo, Qingdao, China). The laboratory in which the tests were conducted was maintained at a constant temperature and humidity.

### 2.3. Measuring Methods

#### 2.3.1. 1 $m^3$ Climate Chamber Method

The two pieces (e.g., 1-A1 and 1-A2) with dimensions of 500 × 500 mm of each sample were placed in a formaldehyde balance chamber at a temperature of 23 ± 1 °C and a relative humidity of 50 ± 5%. The distance between the samples was at least 25 mm to allow air to freely circulate throughout the chamber and to contact all surfaces of the samples. After 15 ± 2 days in the chamber, the edges of the samples were sealed with aluminum tape, except for 750 mm of the edge, which was not sealed. Then, the samples were placed in a 1 $m^3$ climate chamber at a temperature of 23 ± 0.5 °C, relative humidity of 50 ± 3%, and air exchange rate of 1.0 $h^{-1}$. The next day, the sampling and measurements began.

#### 2.3.2. Desiccator Method

The dimensions (150 mm × 50 mm × 18 mm) of the samples to be measured using the desiccator method were chosen to ensure that the total surface area was close to 1800 $cm^2$. The samples were conditioned under standard conditions at a temperature of (20 ± 2) °C and a relative humidity of 65 ± 5% until they attained constant mass for 7 days and then started the test. Next, a crystallization dish containing 300 mL distilled water was added, and the samples were stored in the desiccator at 20 °C for 24 h to measure the concentration of formaldehyde in the distilled water. This test was repeated twice, and the average of the two results was reported.

## 3. Results

The formaldehyde emission results from each group of veneered particleboard samples measured by the desiccator method and the 1 $m^3$ climate chamber method are shown in Table 2.

**Table 2.** The measured results of formaldehyde emission from veneered particleboard by the desiccator method and the 1 $m^3$ climate chamber method.

| Sample No. | Desiccator (mg/L) | Climate Chamber (mg/m³) | Samples No. | Desiccator (mg/L) | Climate Chamber (mg/m³) |
|---|---|---|---|---|---|
| 1 | 0.24 | 0.018 | 31 | 0.07 | 0.011 |
| 2 | 0.23 | 0.017 | 32 | 0.21 | 0.012 |
| 3 | 0.64 | 0.020 | 33 | 0.34 | 0.019 |
| 4 | 0.26 | 0.022 | 34 | 0.25 | 0.013 |
| 5 | 0.41 | 0.020 | 35 | 0.30 | 0.015 |
| 6 | 0.45 | 0.020 | 36 | 0.30 | 0.012 |
| 7 | 0.38 | 0.022 | 37 | 0.28 | 0.018 |
| 8 | 0.57 | 0.032 | 38 | 0.42 | 0.016 |
| 9 | 0.52 | 0.026 | 39 | 0.30 | 0.015 |

**Table 2.** *Cont.*

| Sample No. | Desiccator (mg/L) | Climate Chamber (mg/m³) | Samples No. | Desiccator (mg/L) | Climate Chamber (mg/m³) |
|---|---|---|---|---|---|
| 10 | 0.49 | 0.030 | 40 | 0.39 | 0.018 |
| 11 | 0.50 | 0.029 | 41 | 0.38 | 0.019 |
| 12 | 0.59 | 0.030 | 42 | 0.27 | 0.023 |
| 13 | 0.50 | 0.019 | 43 | 0.47 | 0.015 |
| 14 | 0.60 | 0.021 | 44 | 0.40 | 0.014 |
| 15 | 0.66 | 0.012 | 45 | 0.38 | 0.015 |
| 16 | 0.45 | 0.019 | 46 | 0.47 | 0.021 |
| 17 | 0.52 | 0.013 | 47 | 0.31 | 0.014 |
| 18 | 0.63 | 0.019 | 48 | 0.35 | 0.016 |
| 19 | 0.49 | 0.014 | 49 | 0.02 | 0.006 |
| 20 | 0.70 | 0.020 | 50 | 0.03 | 0.007 |
| 21 | 0.50 | 0.014 | 51 | 0.03 | 0.009 |
| 22 | 0.52 | 0.020 | 52 | 0.02 | 0.011 |
| 23 | 0.61 | 0.020 | 53 | 0.02 | 0.008 |
| 24 | 0.51 | 0.018 | 54 | 0.03 | 0.006 |
| 25 | 0.30 | 0.013 | 55 | 0.02 | 0.007 |
| 26 | 0.26 | 0.010 | 56 | 0.10 | 0.011 |
| 27 | 0.28 | 0.012 | 57 | 0.05 | 0.008 |
| 28 | 0.33 | 0.014 | 58 | 0.02 | 0.010 |
| 29 | 0.42 | 0.021 | 59 | 0.09 | 0.008 |
| 30 | 0.28 | 0.011 | 60 | 0.02 | 0.008 |

## 4. Discussion

From the results provided in Table 1, the correlations between the two methods were analyzed using IBM SPSS Statistics 25 statistical software. The formaldehyde emission data obtained from the desiccator method were plotted on the abscissa, and the formaldehyde emission data obtained from the climate chamber method were plotted on the ordinate. The relationship between the two different measuring methods is depicted in Figure 4.

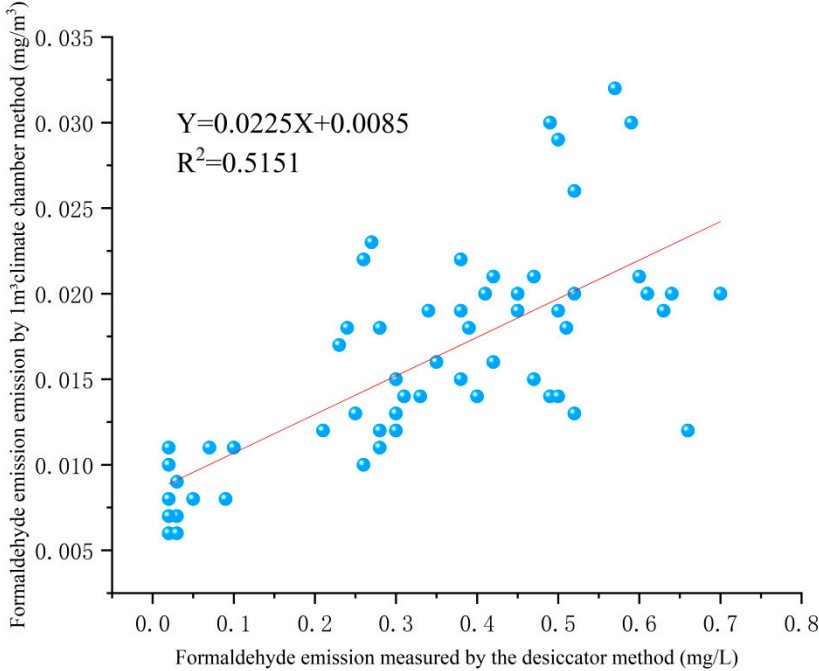

**Figure 4.** The relationship of formaldehyde emissions between the desiccator method and the climate chamber.

Regression analysis of the emission data obtained from the desiccator method and the climate chamber method was performed, and the correlation results are shown in Table 3. The regression model data are shown in Table 4, and the data from the analysis of variance (ANOVA) of the regression model are shown in Table 5.

**Table 3.** Correlation analysis of formaldehyde emissions between the desiccator and climate chamber.

|  |  | **Desiccator** | **Climate Chamber** |
|---|---|---|---|
|  | Pearson correlation | 1 | 0.718 ** |
| Desiccator | Sig.(2-tailed) | / | 0.000 |
|  | Number of cases | 60 | 60 |
|  | Pearson correlation | 0.718 ** | 1 |
| Climate Chamber | Sig.(2-tailed) | 0.000 | / |
|  | Number of cases | 60 | 60 |

** At 0.01 (2-tailed), with a high correlation.

**Table 4.** Regression model of formaldehyde emissions between the desiccator and climate chamber.

| **Model** [a] | | **Unstandardized Coefficients** | | **Standardized Coefficients** | *t* | **Significance** |
|---|---|---|---|---|---|---|
|  |  | **B** | **Standard Error** | **Beta** |  |  |
| 1 | (Constant) | 0.009 | 0.001 | / | 7.623 | 0.000 |
|  | Desiccator | 0.023 | 0.003 | 0.718 | 7.849 | 0.000 |

[a] Dependent Variable: 1 m$^3$ Climate Chamber.

**Table 5.** Variance analysis of regression model of formaldehyde emissions between desiccator and climate chamber.

| **Model** | **DOF** | **Sum of Squares** | **Mean Square** | **F** | *p* |
|---|---|---|---|---|---|
| Regression | 1 | 0.001 | 0.001 | 61.609 | ≤0.001 |
| Residual | 58 | 0.001 | 0 | / | / |
| In total | 59 | 0.002 | / | / | / |

As shown in Tables 2–4, the Pearson's linear correlation coefficient of the two methods (R) was 0.718, indicating that they were correlated. The F = 61.609 and $p \leq 0.001$ indicated that the regression equation of the fitted data was extremely significant when the confidence level was a = 0.01, and the data were linearly correlated. For the 60 groups of test data, the established regression model is shown in Table 3. Upon fitting the data, the regression equation of the correlation model was y = 0.0225x + 0.0085, and the coefficient of correlation ($R^2$) was 0.5151. This means that 51% of the variation is explained by this model of regression, and the other 49% of the variation is due to other factors or random variation.

Xiaorong L. and Xianyuan L. selected 20 wood-based panel samples with different specifications and models as the research object, and the formaldehyde emission of each sample was tested by the 1 m$^3$ climate chamber method and desiccator method, respectively. Through comparative analysis, it is found that the formaldehyde emission of MDF, particleboard, and plywood samples in the experiment varies in the range of 0.5–0.7 mg/L when the test value of the climate chamber method is 0.100–0.124 mg/m$^3$. When the concentration of the desiccator method reaches 0.7 mg/L, the concentration of the climate chamber method exceeds the limit requirement of ≤0.124 mg/m$^3$ [18]. The results reflect the correlation between the two formaldehyde measurement methods, but this experiment only selected 20 group samples to test in the laboratory; the experimental data are not extensive enough.

This research is combined with the specific situation in the production process of enterprises, selects the commonly used panels of enterprises as samples, carries out 60 groups of testing experiments, analyzes its correlation through more experimental data, and the reliability of the results is higher.

## 5. Conclusions

In this study, the desiccator and 1 m³ climate chamber methods of GB/T17657-2013 were used to measure formaldehyde emissions from veneered particleboard. Statistical analysis of the emission data indicated that there was a linear relationship between the two methods. The linear regression equation upon fitting of the data was y = 0.0225x + 0.0085, and its coefficient of correlation ($R^2$) was 0.5151. The model, therefore, could explain 51% of the variation in the data using the two methods. However, the measurements were affected by many uncertain factors. Although the fitness of the linearity was low, regression analysis has an inherently low universal applicability. The methods in this study can be referenced to establish a linear regression model between different methods. The climate chamber method is not economically feasible for many factories; thus, a correlation with other methods is necessary. The established regression model makes it possible to use the desiccator method for routine control in factories, but only in the case of veneered particleboards. Future research is needed on other types of panels.

**Author Contributions:** Conceptualization, Z.Z.; methodology, J.S.; validation, L.D. and T.Y., data curation, Y.C.; writing—original draft preparation, J.S.; writing—review and editing, J.Z.; visualization, Y.C.; supervision, J.S.; project administration, J.S.; funding acquisition, J.Z. All authors have read and agreed to the published version of the manuscript.

**Funding:** This research was funded by the Chinese National Promotion Program of Forestry and Grassland Scientific and Technological Achievements, grant number 2020133139 and The Oppein Home Group INC, grant number 2020.

**Institutional Review Board Statement:** Not applicable.

**Informed Consent Statement:** Not applicable.

**Data Availability Statement:** Not applicable.

**Conflicts of Interest:** The authors declare no conflict of interest.

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
