# Peer review of "Correlation between the Desiccator Method and 1 m³ Climate Chamber Method for Measuring Formaldehyde Emissions from Veneered Particleboard"

_processes, doi:10.3390/pr10051023_

Round 1
Reviewer 1 Report
The manuscript is focused on experimental investigation of the formaldehyde emissions, released from wood-based composite (particleboard), and a comparison between two of the most common measuring methods, i.e. desiccator method and chamber method.
Please, see below my comments on your work:
In general, the title (lines 2-4), abstract (lines 10 to 24) and the keywords (lines 25-26) correspond to the aims and objectives of the manuscript.
Line 4, title of the manuscript: “particleboard” is a one word, please revise throughout the manuscript.
The abstract is concise, specific, and outlines the aim and main findings of the research.
In the keywords I would recommend to add also “wood-based panels”.
Line 31: “glued” should be replaced by “bonded”, please revise.
Lines 34-35: the statement that formaldehyde is hazardous is true, however, I’d recommend to add additional information on the human health-related hazards associated with formaldehyde. In addition, in line 35 it should be mentioned that “…it is carcinogenic to humans”.
In the Introduction, I’d recommend to add a short paragraph about the factors (both internal and external) affecting the formaldehyde emission from wood composites. Please check this relevant references on the topic:
https://doi.org/10.1155/2018/9349721
In addition to the Chinese standards described, I’d strongly recommend to add a section comparing the different standards for determining formaldehyde emission from wood-based panels in China, USA, Europe, and Japan. It would be very useful from a practical point of view. Please check the following relevant references:
https://doi.org/10.1080/17480272.2022.2056080
https://doi.org/10.1159/000024655
In general, the Introduction section is informative, but should be extended based on the recommendations given above. This will significantly increase the scientific soundness of the paper.
Line 118: the dimensions of the particleboards used should be given like this: 1220 mm x 2440 mm x 18 mm, please revise.
Line 131: Figure 1 is of very bad quality and it is very hard to read it, please replace it.
Line 152: Figures 2 and 3 - the same comment as above, please provide a better quality figure. In line 152 the caption of the left figure should be Figure 2, not Figure 1, please revise.
Lines 158-163: please provide relevant information on the testing equipment used, i.e. company producer, city, country. Please add information about the ambient temperature and relative humidity at which the tests were performed.
Please explain why did you use different temperature and relative humidity values for carrying out the desiccator and chamber measurements.
Please add relevant information on the statistical software used in your research.
Line 244: Figure 4 is not of good quality, please replace it.
There is absolutely no discussion of the results obtained, given in Figure 4 and tables 2-4, which is a serious flaw of the manuscript. Please compare and discuss your results with previous research works in the field.
Overall, the Conclusion part (lines 281-293) reflects the main findings of the manuscript.
The References cited are appropriate to the research topic, but their number (only 10) is very low. Please add more references, especially in the Introduction and Discussion sections of your manuscript, as recommended above.
The references are not properly formatted in accordance with the journal requirements, please check the Instructions for Authors.
Best regards!
Reviewer 2 Report
It is a good work. There are some passages that need to be reformulated and there are missing data. Please, check and modify according to the suggestions in the Report. There are few references. Please, improve your introduction and discussion chapters with new references.

Reviewer 3 Report
Dear Authors,
apart from several remarks, mentioned below, I found your manuscript highly valuable for industrial readers, looking for simplification of measurements of formaldehyde emission. However, please, consider the following remarks:
- line 40, you listed several methods of formaldehyde emission, including perforated extraction method; if you said about the method according to EN 120 standard, please, consider, that this is the method of measurement of formaldehyde content, and not emission/release; please, make the references to this method properly (since it is not applied to measure formaldehyde emission)
- line 66-69: you said "Qionghui Zhao (2018) measured the formaldehyde emission of blockboard by two different methods, the desiccator method and 1 m3 climate chamber method, and compared the results to elucidate the influence of the different measurement methods on the formaldehyde emissions." I think there is no influence of the measurement method on the formaldehyde emission, rather than the measurement method can influence the results of measurement of formaldehyde emission.
- line 107-114 - you provided there a kind of the aim of the paper, mentioning that you consider the influence of certain parameters [...] on the reliability of formaldehyde emissions from the wood-based panels. In fact, the results chapter has shown the comparison of results of formaldehyde emission, achieved by two methods. There are no results of the "influence of certain parameters" (as you mentioned in lines 107-110). Please re-write lines 107-113 to get closer to the results and conclusions presented.
Best regards!
Round 2
Reviewer 1 Report
Although the authors have revised the manuscript and most of my previous comments have been properly addressed, I still miss one major remark in the updated version, i.e. a comparative data on the different standards used for the determining the formaldehyde emission from wood composites, as well as the respective limit values, preferably in a tabe form. This can make the paper more useful for the industrial practice. Please check the following relevant paper: https://doi.org/10.1080/17480272.2022.2056080
